# Curtailing Lead Aerosols: Effects of Primary Prevention on Declining Soil Lead and Children’s Blood Lead in Metropolitan New Orleans

**DOI:** 10.3390/ijerph16122068

**Published:** 2019-06-12

**Authors:** Howard W. Mielke, Christopher R. Gonzales, Eric T. Powell

**Affiliations:** 1Department of Pharmacology, Tulane School of Medicine, 1430 Tulane Ave. 8683, New Orleans, LA 70112, USA; chrisgc99@gmail.com; 2Lead Lab. Inc. New Orleans, LA 70119, USA; powellet2@gmail.com

**Keywords:** blood lead decline, Cochrane collaboration, soil lead decline, mapping soil lead and blood lead, tetraethyl lead, topsoil

## Abstract

After decades of accumulation of lead aerosols in cities from additives in gasoline, in 1975 catalytic converters (which are ruined by lead) became mandatory on all new cars. By 1 January 1986 the rapid phase-down banned most lead additives. The study objective is to review temporal changes of environmental lead and children’s blood lead in communities of metropolitan New Orleans. In 2001, a soil lead survey of 287 census tracts of metropolitan New Orleans was completed. In August–September 2005 Hurricanes Katrina and Rita storm surges flooded parts of the city with sediment-loaded water. In April–June 2006, 46/287 (16%) of the original census tracts were selected for resurvey. A third survey of 44/46 (15%) census tracts was completed in 2017. The census tract median soil lead and children’s median blood lead decreased across surveys in both flooded and unflooded areas. By curtailing a major urban source of lead aerosols, children’s lead exposure diminished, lead loading of soil decreased, and topsoil lead declined. Curtailing lead aerosols is essential for primary prevention. For the sake of children’s and ultimately societal health and welfare, the long-term habitability of cities requires terminating all remaining lead aerosols and cleanup of legacy-lead that persists in older inner-city communities.

## 1. Introduction

In 2014, the World Health Organization reported that 54.6% of the world population lived in cities, up from 34% in 1960 [1]. Urban environments require systematic, longitudinal research to ensure human safety and environmental sustainability. Long-term studies are often conducted to comprehend the impact of cities on ecosystems. However, studies on the contribution of policy on the resilience and sustainability of cities as human habitats are less common [2]. Lead (Pb) dust is an imperceptible and persistent contaminant. Because of the small particle size (<2.5 µm), Pb aerosol exposure is a major risk to all cells in the body and the cause of many ailments, including chronic neurotoxic and cardiovascular diseases in every age group [3,4,5,6,7,8].

In 1980, the National Academy of Sciences published a report, *Lead in the Human Environment,* that included an assertion by Clair Patterson, “Sometime in the near future it probably will be shown that the older urban areas of the United States have been rendered more or less uninhabitable by the millions of tons of poisonous industrial lead residues that have accumulated in cities during the past century. Babies are more susceptible…than are adults [9]”.

Patterson’s apprehensions about Pb contamination were confirmed a few years later by an urban soil study which reported an extreme disparity between the soil lead (SPb) in the inner-city compared to outer communities of metropolitan Baltimore, Maryland [10]. Subsequent studies in residential communities of Minnesota showed a strong association between SPb and children’s blood Pb (BPb) in the context of the location within cities and between various sized cities [11].

The gasoline additive tetraethyl lead (TEL) was an especially potent source of Pb contamination of cities. TEL was introduced into commerce in the mid-1920s [12]. Its use became almost universal by 1950. The Pb tonnages from vehicle use grew exponentially through the 1950s, ‘60s, ‘70s, and peaked in 1975 [13]. An estimate of quantities of aerosol Pb from TEL in the U.S. was 6 million tonnes (metric) with about 10,000 tonnes deposited in New Orleans [14]. The aerosol inputs of Pb-dust varied with traffic flow and congestion resulting in the automobile as a toxic substance delivery system. The traffic-associated TEL caused a public health disaster that became international in scope [15,16].

In 1975 catalytic converters became mandatory on all new cars. TEL usage declined steadily nationally and locally after the introduction of catalytic converters, a pollution control device that necessitated the removal of Pb additives to prevent damage to the catalyst [17]. By 1 January 1986 the rapid phase-down banned most lead additives (see Figure 1) [13]. Reducing TEL contributed to a major advance in primary prevention. 

TEL is a still a legal product in the U.S. and used as an additive to aviation gas or LL100 avgas (containing 0.56 g Pb per L). The U.S. EPA estimates that avgas now accounts for about 60% of the current Pb aerosol in the U.S. [18,19]. An outcome of avgas is that BPb is higher for children living within 0.5–1 km of airports where avgas is used compared with children living 1 km beyond these airports [20]. Particularly concerning is the fact that all grades of fuel, leaded and unleaded gasoline, are transported through the same pipelines. To protect petroleum industry from liability due to lead contamination of fuel an allowable amount of TEL is permitted in unleaded gasoline.

Current U.S. intervention efforts take place only after individual children are identified with elevated BPb and this approach is a disservice to children because it is secondary, not primary prevention [21]. Cochrane Collaboration is internationally recognized for evaluating the effectiveness of medical interventions [22]. Cochrane Collaboration reviewed the outcomes of U.S. Pb interventions methods which involve education and housecleaning after children are identified as Pb exposed. The review indicated that education and household cleanup methods are *ineffective* for treating Pb exposed children. The review also noted that the effect of SPb remediation and/or a combination of actions on children’s exposure were not reviewed because not enough data exists [22].

The primary objectives of this study are to fill the knowledge gap about the linkages between Pb aerosols, soil Pb, and childhood Pb exposure, and to review changes in soil Pb and children’s blood Pb in metropolitan New Orleans 20 to 30 years after the cessation of most Pb aerosols from TEL additives in vehicle fuels.

## 2. Methods

The natural experimental design began serendipitously with an initial SPb survey of 287 census tracts in metropolitan New Orleans. The survey was completed in 2001, five years (4 years, 8 months) before Hurricane Katrina on 29 August 2005 [23]. As described in the blood lead section below, BPb was collected independently by the Louisiana Office of Public Health. To evaluate the effect of the hurricanes, a second, post-Katrina SPb survey of 46 census tracts was completed 9 months after Hurricane Katrina, 8 months after Hurricane Rita [24]. The third survey was completed in 2017 and included 44/46 census tracts [25]. The three surveys were conducted within the 1990 census tract boundaries [26]. The word “community” is used interchangeably with “census tracts.” Figure 2 shows the locations and depth of flooding of the 44 census tracts that are the topic of this study.

### 2.1. Soil Collection

The soil samples were collected from the top 2–3 cm of 44 census tracts. For each census tract, 19 soil samples from 4 residential locations were systematically collected as follows: Within 1 m of residential roadsides (9 samples), within 1 m of busy streets (4 samples), within 1 m of homes (3 samples), and from open spaces away from homes and streets (3 samples) [23].

### 2.2. Soil Pb Extraction and Analysis

The collected soils were air-dried and sieved through a 2 mm mesh sieve (ASTM#10 stainless steel sieve). The protocol for extraction was tailored to manage large numbers of samples with maximum efficiency using minimum dilution and sample re-analysis in the following manner: To overcome the alkalinity and dilution issues, soil samples were extracted with 1 M nitric acid at a 1/50 ratio (0.4 g soil/20 mL 1 M nitric acid), and shaken for two hours [23]. The soil to acid ratio maintained a low pH for extraction and decreased the need for sample dilution and reanalysis. The number of soil sample results used in this study were 829, 836, and 836 for the 1998–2001 (pre-Katrina), 2006, and the 2013–2017 surveys, respectively (see Table 1).

### 2.3. Soil Analysis Quality Assurance Quality Control (QAQC)

National Institute of Standards and Technology (NIST) traceable standards were used for ICP-AES calibration. Duplicate soil samples at a rate of 1 per census tract were prepared and analyzed. In-house reference soil samples were included during analysis, and the SPb results for these were consistent across the three surveys. All survey soil samples are archived at Tulane University.

### 2.4. Children’s Blood-lead Data

Blood Pb biomonitoring results were collected by the Louisiana Office of Public Health’s Louisiana Healthy Homes and Childhood Lead Poisoning Prevention Program (LHHCLPPP). The children’s BPb biomonitoring program follows protocols, established by the Centers for Disease Control and Prevention, for collection, preparation, and analysis of children’s BPb [27,28]. For each survey ≤6-year-old children’s blood samples were collected by clinics throughout metropolitan New Orleans and the BPb results were relayed to the LHHCLPPP. The BPb data (unidentifiable as to individual children) were obtained from LHHCLPPP and coded by census tracts. The BPb results are given in μg/dL. An elevated BPb is defined as equal to or above the 2012 reference value of 5 μg/dL [21]. The pre-Katrina BPb results are from January 2000–August 2005, the second survey results are from 2006–2010, and the third survey results are from 2011–2016. The BPb data for metropolitan New Orleans consists of 10,554, 4723, and 5315, respectively for the three surveys.

### 2.5. Statistical Analysis

The Multi-Response Permutation Procedures (MRPP) are a group of distance-based statistical tests [29,30]. The statistical tests evolved from the work of R.A. Fisher [31,32]. The model does not assume any specific data distribution and the statistical model focuses on the actual data (without transformations, truncation, or other manipulation to “normalize” the data). Furthermore, the model treats data using ordinary Euclidian geometric spaces. The probability value (*p*-value) associated with the MRPP is the proportion of all possible test statistic values under the null hypothesis that are less than or equal to the observed test statistic of the actual observations.

## 3. Results and Discussion

Table 1 shows percentiles of the median SPb and BPb results in three sets of 44 census tracts in metropolitan New Orleans.

Table 2 shows the statistical results of the MRPP analysis for SPb (left panel) and BPb (right panel) from the three surveys of 44 communities in Metropolitan New Orleans. The decreases in BPb are particularly remarkable. According to the results (*p*-value = 1.09 × 10^−22^), chance alone does not play a role in the decreases of BPb across the three surveys. The results for SPb are less extreme (*p*-value = 0.011), but they still indicate a substantial reduction of SPb across the three surveys. The largest decreases occur between the first survey compared with the second and third surveys.

### Temporal Decreases in SPb and BPb during Three Surveys of 44 Communities

Figure 3 is a graph of median soil Pb vs. median blood Pb for the 44 census tracts in this study. Comparing the surveys, there were consistent reductions of children’s BPb in the 44 communities surveyed in 1998–2000 (red) vs. 2006–2010 (blue) and 2011–2017 (green). The reductions of BPb appear related to reductions of SPb in 44 communities and reflect the extraordinary changes in the environment of New Orleans after the removal of TEL from gasoline. Comparable SPb and BPb decreases occurred in unflooded and flooded communities, and thus, flooding alone was not the only process that decreased SPb and BPb in metropolitan New Orleans.

The removal of TEL from highway vehicle gasoline curtailed most of the Pb-dust inputs into cities. The sharp decrease of Pb aerosols had the immediate effect of reducing children’s inhalation of Pb particles and rapidly reducing BPb [33,34,35]. When Pb aerosols declined, Pb deposition or loading of the topsoil also decreased. Contaminated soil remaining in residential communities persists as a source of Pb exposure. On a seasonal basis, especially during late summer and fall, soil moisture becomes depleted and soil particles become prone to resuspension into the air, accounting for BPb seasonality [36,37]. During winter months, BPb trends higher and during late summer and fall BPb trends lower [38,39,40]. After Pb aerosol deposition ceased, soils had a reprieve. Pb particles infiltrate into deeper soil horizons, accounting for a portion of the decrease of Pb in topsoil of metropolitan New Orleans communities. Support for this process is found in a study in Israel and also shows up in the three-decades of research in the Vienna Woods where declining atmospheric deposition of metals are reflected by decreases in soil and foliage at the study site [41,42].

Topsoil ecosystems are teaming with lifeforms [43]. The literature on bioturbation and biogeomorphology is rich including early to current research, and these processes may be critical for explaining topsoil Pb decreases in flooded as well as unflooded New Orleans communities [44,45]. The influences of physical and biological processes on urban topsoil require further research to understand the ecological contribution of organisms to the resilience and sustainability of urban soils [2].

## 4. Continuing Exposure and Primary Lead Prevention for Children

Although there have been remarkable decreases of SPb and BPb throughout both previously flooded, as well as unflooded communities, many inner-city communities remain excessively contaminated from previous Pb uses, and children continue to experience excessive exposure to SPb [14]. Studies in New Orleans demonstrate the dire long-term societal effects of Pb exposure [46,47,48]. Our current understanding supports Patterson’s prediction that because of the excessive residues from industrial use of Pb, cities became too contaminated for the safety for human habitation. Based on the persistence of lead, without active cleanup, SPb will continue to expose children for decades in many older inner-city communities. Several soil projects are underway to diminish urban Pb as a proactive, primary prevention method to reduce children’s BPb [49,50,51,52].

## 5. Conclusions

Our study provides evidence that Pb aerosols created a legacy of accumulation that extends to urban soils outside of homes within residential communities. Furthermore, the amounts of Pb in soils of each community are associated with children’s blood Pb living in the same community. In New Orleans, the amount of lead within communities is declining in both topsoils and children’s blood. The decline observed in New Orleans is probably occurring in other cities. The use of all TEL additives must be banned to prevent additional Pb loading, especially in cities where most of the human population resides. Some communities contain soils that remain excessively lead contaminated, a situation that requires primary prevention to thwart the disproportionately high lead exposure of many children.

## Figures and Tables

**Figure 1 ijerph-16-02068-f001:**
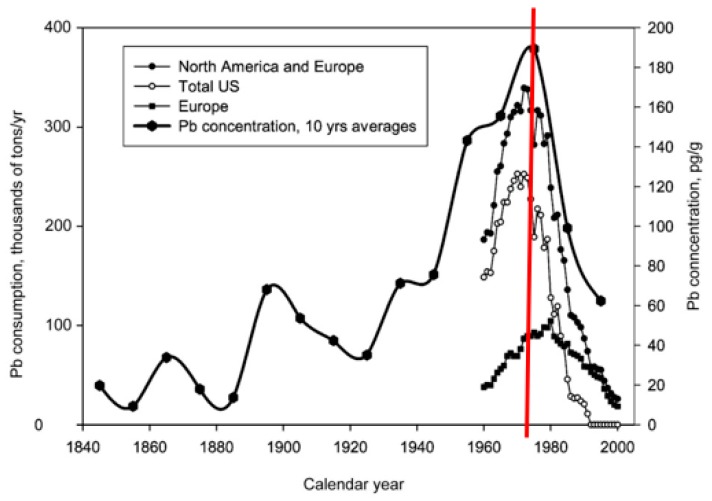
Rise and fall of lead aerosol deposition in Arctic ice before and after the 1975 introduction of unleaded gasoline to protect the catalytic converter, modified from [13].

**Figure 2 ijerph-16-02068-f002:**
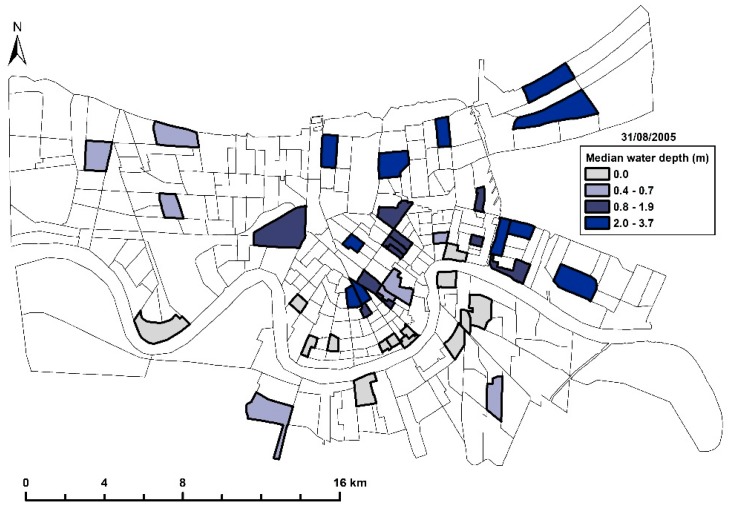
Map of the 44 census tracts in this study showing median flood water depth from Hurricane Katrina (31 August 2005).

**Figure 3 ijerph-16-02068-f003:**
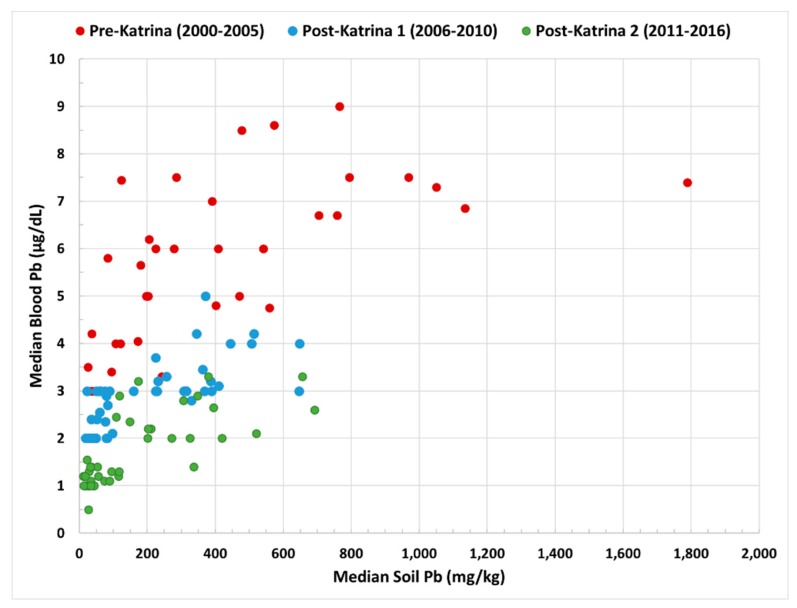
The Y axis shows the median children’s blood Pb in 44 communities for 1998–2000 (red) compared with communities surveyed in 2006–2010 (blue) and 2011–2017 (green). For each community children’s median blood Pb are paired in the X axis with the corresponding median soil Pb.

**Table 1 ijerph-16-02068-t001:** Percentiles of soil Pb (SPb) and children’s blood Pb (BPb) median results of 44 residential communities in metropolitan New Orleans. Note the continuous 50% (bold) SPb decreases (left panel) from 1998–2000 through 2013–2017, and the continuous 50% (bold) BPb declines (right panel) from 2000–2005 through 2011–2016.

	SPb (mg/kg)			BPb (µg/dL)		
	1998–2000	2006	2013–2016	2000–2005	2006–2010	2011–2016
Survey code	SPb2	SPb3	SPb4	BPb2	BPb3	BPb4
N CT	44	44	44	44	44	44
N SPb & BPb	829	836	836	10,554	4723	5315
min	25	19	11	2.0	2.0	0.5
5%	26	23	13	3.0	2.0	1.0
10%	32	35	17	3.0	2.0	1.0
25%	45	53	27	3.0	2.4	1.0
**50%**	**200**	**129**	**82**	**5.0**	**3.0**	**1.4**
75%	525	359	256	6.8	3.2	2.2
90%	881	475	407	7.5	4.0	2.9
95%	1114	613	622	8.6	4.2	3.3
max	1789	647	692	9.0	5.0	3.3

**Table 2 ijerph-16-02068-t002:** Multi-Response Permutation Procedure Analysis of Blood Pb (left panel) and Soil Pb (right panel) results in 44 communities over three surveys. Notice that the extreme *p*-value of Blood Pb differences indicating that the decline has been much larger than the decline observed for Soil Pb.

Blood Pb			Soil Pb		
Survey Code	Group Size	Group Distance	Survey Code	Group Size	Group Distance
BPb2	44	2.22	SPb2	44	383.7
BPb3	44	0.78	SPb3	44	198.3
BPb4	44	0.84	SPb4	44	181.5
**Results**			**Results**		
Delta Observed = 1.28			Delta Observed = 254.47		
Delta Expected = 2.04			Delta Expected = 262.47		
Delta Variance = 0.00034			Delta Variance = 5.61		
Delta Skewness = −1.61			Delta Skewness = −1.72		
Standardized test statistic = −41.39			Standardized test statistic = −3.37		
Probability (Pearson Type III) of a			Probability (Pearson Type III) of a		
smaller or equal Delta = 1.09 × 10^−22^			smaller or equal Delta = 0.011		

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
