# Peer review of "Curtailing Lead Aerosols: Effects of Primary Prevention on Declining Soil Lead and Children’s Blood Lead in Metropolitan New Orleans"

_ijerph, 2019, doi:10.3390/ijerph16122068_

Round 1
Reviewer 1 Report
Thank you for the opportunity to review this paper. This research is an important follow-up to the soil lead work in New Orleans. Here are my suggestions:
Abstract - Lines 18/19: might be helpful to mention how hurricanes impact soil lead levels
Lines 167-170: Need to expand up this topic or remove it from the paper.
Could there possibly be other sources of lead in addition to aerosols in the city - for example, foundry sand or other dumping?
An explanation of how hurricanes remove soil lead (although it might be obvious) is needed.
Author Response
Thank you for the opportunity to review this paper. This research is an important follow-up to the soil lead work in New Orleans. Here are my suggestions:
Abstract - Lines 18/19: might be helpful to mention how hurricanes impact soil lead levels
"In August-September 2005 Hurricanes Katrina and Rita storm surges flooded parts of the city with sediment-loaded water. In April-June 2006, 46/287 (16%) of the original census tracts were selected for resurvey. A third survey of 44/46 (15%) census tracts was completed in 2017. The census tract median soil lead and children’s median blood lead decreased across surveys in both flooded and unflooded areas."
Lines 167-170: Need to expand up this topic or remove it from the paper.
"Topsoil ecosystems are teaming with lifeforms [45]. The literature on bioturbation and biogeomorphology is rich including early to current research, and these processes may be critical for explaining topsoil Pb decreases in flooded as well as unflooded New Orleans communities [46, 47]. The influences of physical and biological processes on urban topsoil require further research to understand the ecological contribution of organisms to the resilience and sustainability of urban soils [2].
Could there possibly be other sources of lead in addition to aerosols in the city - for example, foundry sand or other dumping? We looked closely at multiples sources of lead in soils. Incinerators and incinerator ash disposal sites were considered but they did not appear to affect the major pattern of lead in New Orleans. There was a smelter on the West Bank and it did affect the community. But this was not included in this examination of 44 census tracts of New Orleans.
An explanation of how hurricanes remove soil lead (although it might be obvious) is needed. See comments about topsoil ecosystems above. We note that similar changes are apparent in both flooded and unflooded communities. Thus, also in the conclusions we added: "Although there have been remarkable decreases of SPb and BPb throughout both previously flooded, as well as unflooded communities, many inner-city communities remain excessively contaminated from previous Pb uses, and children continue to experience excessive exposure to SPb [14]."
Reviewer 2 Report
This is an excellent paper, well-written and adds to the literature about SPb and BPb.
Table 1 might be easier to interpret as a chart of figure.
There's an error in line 144. It reads Figure 2 and it should be Figure 3.
The authors could add a stronger statement in the conclusion about the need to ban all TEL.
Author Response
This is an excellent paper, well-written and adds to the literature about SPb and BPb.
Table 1 might be easier to interpret as a chart of figure. We tried many alternative charts or figures and the only way they were readable was when data was removed.
There's an error in line 144. It reads Figure 2 and it should be Figure 3. Excellent, this is fixed.
The authors could add a stronger statement in the conclusion about the need to ban all TEL. Line 185. "The use of all TEL additives must be banned to prevent additional Pb loading of the environment."